# Introduction to Door Opening Type Classification Based on Human Demonstration

**DOI:** 10.3390/s23063093

**Published:** 2023-03-14

**Authors:** Valentin Šimundić, Matej Džijan, Petra Pejić, Robert Cupec

**Affiliations:** Faculty of Electrical Engineering, Computer Science and Information Technology Osijek, 31000 Osijek, Croatia

**Keywords:** door handles, hidden handles, push mechanism, door opening type, human demonstration, robot vision

## Abstract

Opening doors and drawers will be an important ability for future service robots used in domestic and industrial environments. However, in recent years, the methods for opening doors and drawers have become more diverse and difficult for robots to determine and manipulate. We can divide doors into three distinct handling types: regular handles, hidden handles, and push mechanisms. While extensive research has been done on the detection and handling of regular handles, the other types of handling have not been explored as much. In this paper, we set out to classify the types of cabinet door handling types. To this end, we collect and label a dataset consisting of RGB-D images of cabinets in their natural environment. As part of the dataset, we provide images of humans demonstrating the handling of these doors. We detect the poses of human hands and then train a classifier to determine the type of cabinet door handling. With this research, we hope to provide a starting point for exploring the different types of cabinet door openings in real-world environments.

## 1. Introduction

The ability to open doors is important for robots designed for use in household environments, as it allows them to reach places behind doors, which is necessary for solving many tasks. Doors are a common feature in most homes, and robots designed to perform tasks or provide assistance in a home environment must be able to navigate through doors to get from one room to another. Similarly, the ability of robots to open cabinets and drawers can also be important in a home. Cabinets and drawers are commonly used to store food, cleaning supplies, tools, and other household items. A robot that is designed to perform tasks such as retrieving items, replenishing supplies, or organizing items may need to be able to open cabinets to access these items and perform its tasks. For example, a robot designed to assist with cooking may need to be able to open cabinets to pick ingredients or cooking utensils. Similarly, a robot designed to help with cleaning might be able to open cabinets to access cleaning supplies or tools.

Traditionally, cabinets and drawers are opened by grabbing and pulling the handle that is attached to the front of this furniture. Therefore, many researchers have already developed various algorithms for handle detection and grasp planning. However, in modern homes, it has become increasingly common to see cabinet and drawer doors without visible handles on their outer surface. These types of doors are usually opened by pressing on the outer surface near a spring mechanism located inside the door, as shown in Figure 1b. When this spring, shown in Figure 1a, is pressed, it stretches and pushes the door from the inside, allowing it to be opened. On the other hand, there are also cabinet doors that are opened by grasping the edges of the front surface of the door, as shown in Figure 1c, which are referred to in this paper as hidden handle doors.

These opening types can present a potential problem for autonomous robots that are programmed to open doors, as these algorithms typically expect to find a visible handle suitable for grasping. Cabinet doors without visible handles or with hidden handles may be more difficult for these robots to open, as they do not have a traditional handle to grasp and use to initiate the opening process. To the best of our knowledge, there is no recent research considering opening doors without regular handles.

In this research paper, we focus on the first step of the process of opening doors and drawers with regular and alternative handles: the classification of the opening type. To conduct our study, we collected and labeled a dataset of RGB-D images of various cabinet and drawer doors in their natural environment, such as kitchens, living rooms, and bedrooms. These images were taken in complex scenes to more accurately represent the real-world situations in which these doors are typically encountered.

One of the challenges we identified when attempting to classify the opening type of these doors is that when the regular handle is not present on the door surface, it can be difficult to determine the correct method for opening the door. In some cases, doors with hidden handles may resemble those with a push mechanism, making it difficult to distinguish between the two. As a result, we decided to train an algorithm to distinguish between these opening types based on human demonstration. When a human opens a door with a push mechanism, they typically press the door surface with their spread palm. On the other hand, when a human opens a door with a hidden handle, their fingers usually bend to grip the edge of the surface. By using these human demonstration patterns as examples, we develop an algorithm that can classify the opening type of these doors.

The proposed approach consists of image preprocessing, classification training, and evaluation. Since the images are acquired in complex scenes, the first step consists of detecting the region of interest (ROI) on the image represented by the considered bounding box of the door. Using this ROI as input to the OpenPose [1] and ResNet [2] algorithms, we obtained door opening classes. This training was done with images containing human hands to demonstrate opening and with images without hands. The idea was to find out if the human demonstration could help to identify the type of door opening. This can be useful in cases where a robot deployed in a work environment cannot correctly determine the opening type of a particular door. A human could then show the robot how to open that door by demonstration.

We propose the following contributions in this paper:HoDoor: An RGB-D dataset of cabinets and drawers in complex scenes with labeled bounding boxes and ground truth information about the type of door opening. Images are provided from the same viewpoint with and without a human hand to demonstrate opening.A neural network-based approach to classify three door opening types: with a regular handle, with a hidden handle, and with a push mechanism.

The paper consists of the following sections. In Section 2, the related research is presented. We researched the state of the art in the field of different detection methods that could be used for door and handle detection. Furthermore, we explored hand pose estimation and hand gesture recognition in images, since our method is based on human door opening demonstrations. Section 3 provides information about the HoDoor dataset and proposed methodology of door opening type classifications. Results are given in Section 4, and they are discussed in Section 5.

## 2. Related Research

### 2.1. Object Detection

Object recognition is a popular research area, and there is an extensive body of literature on it with broad reviews of different methods and approaches [3,4,5,6,7]. While early approaches to object recognition relied on feature extraction combined with traditional machine learning methods to search for regions of interest (ROIs), modern approaches generally use deep neural networks capable of extracting and learning features directly from data. Some popular methods for object detection are YOLO and its variants [8,9,10,11,12], Faster R-CNN [13], Single Shot MultiBox Detector (SSD) [14], Mask R-CNN [15], and many others. Object detection can be considered a combination of image classification and object localization. Image classification can also be used independently of object detection with an ROI or an entire image. Deep learning based methods have achieved great success in image classification [16]. Some popular methods for image classification are: VGG [17], Inception [18], EfficientNet [19], MobileNet [20], etc. In this work, we use ResNet [2] to classify the type of door opening based on the door ROI obtained from the YOLO detector.

The You Only Look Once (YOLO) algorithm [8] is a single-shot, real-time object detection algorithm that converts classification tasks into a regression problem. This is achieved by separating the bounding boxes prediction and the associated class probabilities. Since then, several updates to the basic YOLO algorithm have been published [9,10,11,12]. This work uses the latest YOLO variant and the state-of-the-art detection algorithm YOLOv7 [12] for the problem of detecting doors in the image containing objects with doors. YOLOv7 is based on the YOLOv4 [11] algorithm and improves accuracy and inference times by introducing several structural and optimization modifications: the Extended Efficient Layer Aggregation Network (E-ELAN), which is an architecture in the backbone of the framework that allows the framework to continuously learn from the data while not disturbing the existing flow of gradients during training. Additionally, the compound scaling method was proposed, which subsequently scales the width of the block with the same amount of change to the transition layers. This results in preserving the properties of the initial model design and the optimal structure. Another feature introduced is module-level network reparameterization, which allows network modules to independently readjust weights with gradient flow propagation paths. The final model improves the inference results. The proposed YOLOv7 achieves the highest accuracy of 56.8% AP on the COCO dataset with 30 FPS or higher on GPU V100 [12].

#### Door and Handle Detection

Door detection is a common problem in both computer vision and robotics. Extensive research has been conducted on this topic, and the proposed methods differ in their approach. An overview of the approaches for detecting windows and doors was given in [21].

There are several approaches that use only an RGB image as input to detect doors. In [22], Chen et al. proposed a convolutional neural network (CNN) that requires only one RGB image to estimate pose of doors. Ramôa et al. [23] proposed multiple 2D and 3D methods for door detection and state classification that can be used in real time on a low-power device. They used PointNet for 3D door state classification and compared multiple neural networks for 2D detection and classification. In recent years, 3D door detection has received more attention than 2D door detection. In [24], the authors used 3D information with PointNet, FastFCN, and FC-HarDNet for door detection and classification on a low-power device. Much of the literature uses a geometry-based approach to door detection without the aid of machine or deep learning. In [25], 3D sensors were used in a door detection algorithm that uses basic structural knowledge about doors and enables part extraction of doors from point clouds based on constraint region growing. Along with Gaussian probability weights, they are combined to create an overall probability measure. To help the visually impaired, a geometry-based approach for detecting doors in closed and half-open states was proposed in [26]. The method extracted corners and edges in an RGB image, which were then grouped based on criteria about the geometry of the door to generate door hypotheses. Vlaminck et al. [27] used depth data to find doors in walls by detecting indentations in the wall. If the doors are on the plane of the wall, the color information is used to detect the doors. In [28], a detection method using only traditional computer vision algorithms was proposed. First, the ground plane equation is estimated by taking the largest number of depth points around one of the detected horizontal lines. Then the edge image is computed, and the line segments parallel to the ground plane’s normal are found using the probabilistic Hough transform. The doors are detected if the found line segments match the constraints set by a created model of the doors. A similar method was proposed in [29]. The authors used a fisheye camera together with a depth camera to extract the floor plane region and detect line segments. Door hypotheses were then found by satisfying the constraints of a created door model and confirmed by finding a similarly colored region in the center of the door hypothesis. A task for opening and navigating through doors using a humanoid robot was introduced in [30]. They proposed two approaches for opening doors. The first approach is similar to previous methods in that it is also based on geometry. The second proposed method involves human-assisted door detection, where the human marks the position of the handle and the normal to the door plane, which then helps in door estimation. All of the above methods achieve good results in terms of accuracy but lack generality, as they can only work with standardized doors for which they have created a model.

To overcome this problem, detectors in the form of neural networks can be used. Recently, YOLO-based approaches have become popular in door detection. Llopart et al. [31] proposed a method that uses RGB-D data for door and handle detection. The approach creates bounding boxes around doors and cabinets using the YOLO detector trained on the ImageNet dataset [32]. After door detection, the method segments handles by color clustering and depth point extraction in the detected ROIs. Another YOLO-based approach was presented in [33], which uses YOLOv3 [10] to detect doors on a mobile robot navigating through a simulation of a building.

In [34], a YOLOv3 trained on a custom dataset was used to detect different types of doors and handles so that a mobile robot could open them. The custom dataset is called DoorDetect Dataset and consists of annotated images taken from the Open Images Dataset containing doors and handles. The authors increased the detail of the information by splitting the class door into three classes. The first class has the same name, “door”, and represents ordinary room doors. The second class includes different types of doors, such as drawers and cabinets, and was named “cabinet door”, while the third class was named “refrigerator door”. The authors used 1013 images for the training phase, while the remaining 200 images were used for the testing phase. They achieved 45% mAP on the test set using the YOLOv3 trained on the custom dataset, compared to 55% and 58% mAP achieved using the YOLOv3 trained on the COCO dataset and VOC 2012 dataset, respectively. The custom DoorDetect dataset has been made publicly available. In this work, the dataset is used to train the YOLOv7 detector, which is then applied to our dataset.

The following research considers handle detection [35,36,37,38,39]. In [36], the authors proposes automation of a door handle cleaning task. The cleaning process included door handle detection based on deep learning, which as a result provided the position of the handle to the robot. In order to improve nursing robots, in [35], identification of doorknobs and handles in images using machine learning was proposed as a preliminary step in the development of an autonomous mobile robot that can open and close arbitrary doors. In [37], the authors propose AI-driven wearable technology, which detects door handles and hand position of the visually impaired user and navigates them to the doorknob. To improve the efficiency of door recognition, in [38], the average height of doorknobs was used as the context feature. Two types of doorknobs were tested: straight (door handle) and round (doorknob). The considered handles in the above-mentioned papers are positioned somewhere on the outer surface of the door and thus visible. To the best of our knowledge, there are no existing methods for recognition of alternative door opening types, such as those with push mechanism and hidden handles, which we propose in this paper.

### 2.2. Hand Pose Estimation

There is extensive research on the topic of hand pose estimation [40,41,42,43,44]. The surveys mostly agree on some problems. The lack of large and generalized datasets causes the methods to perform poorly on complex real-world data and outdoor environments. Another problem is the generalization to hand–object interactions. On the other hand, the surveys have concluded that current methods are usually based on deep learning methods and perform well in estimating the pose of a single hand.

Boukhayma et al. [45] proposed a method that predicts 3D hand pose from a single image. The authors combined a deep convolutional encoder with a model-based decoder. The resulting hand parameters were then reprojected into a 3D hand mesh. Another RGB-based approach [46] uses a graph CNN model that reconstructs a full 3D mesh of the hand. In [47], a cross-platform open-source method was presented that works in two stages. First, a palm detector provides bounding boxes for hands. Then, a hand landmark model is applied to the bounding boxes to predict a 2.5D hand skeleton. A depth-based approach was used in [48] in which the proposed method extracted regions from feature maps generated by a CNN to generate more optimal features for hand poses. The extracted regions were then combined hiearchically based on the features of hand joints. The estimated poses were then iteratively refined to obtain the final result. In [49], an approach was proposed that uses a CNN to estimate a 3D hand pose from a single depth image. The method generates multiple viewpoints of the hand, which are then fed to another neural network to estimate hand poses for each viewpoint. The generated hand poses are then averaged to obtain the final hand pose estimate.

#### Skeleton-Based Hand Gesture Recognition

Hand gesture recognition is a form of interaction between humans and computers. It can be used for interpreting sign language [50,51,52,53], controlling various devices [54,55], gaming and software control [56], and so on.

In skeleton-based approaches, gestures are usually classified based on previously estimated hand poses [57]. In [58], the authors propose a method based on RGB-D data in which the hand joint positions, previously estimated from the depth image, are fed into parallel CNNs. Konstantinidis et al. [59] proposed an LSTM model to classify signs in sign language based on the OpenPose detectors for body and hand pose estimation. In [60], a hand gesture method was proposed where the authors used a support vector machine (SVM) with linear kernels to deal with the high-dimensional representation of a hand pose. In [61], the skeletons are represented as graphs, which allows for spatial relationships between joints to be captured. They are then processed by a spatial-based graph convolutional network (GCN).

In this work, OpenPose [1] was used to estimate hand poses. OpenPose is an open-source system for 2D pose detection. It takes an RGB image and feeds it into a two-branch CNN to predict confidence maps for body parts and the association between them. The predictions are concatenated to obtain better results. In this way, the core of body and foot recognition is formed. Hand detection and face detection are independent blocks that can be turned on or off along with the body detection for each image. Based on the output of the body detector, the bounding box hypotheses are created for the hand with the arm keypoints, which may result in missed detections if the arm keypoints are not detected. The hand keypoint detector can be used independently and is explained in [62]. Simon et al. [62] proposed an RGB-based hand keypoint detection method using a multi-camera setup running in real-time. A keypoint detector was used to provide 2D hand keypoints, which are then reprojected into 3D using a multiview geometry approach, since detection is easier in some views than others. The proposed detector uses a VGG-19 network as a feature extractor on an image patch around the hand. The feature maps are followed by a prediction stage that outputs a set of 21 score maps, each for a different hand keypoint.

## 3. Method

### 3.1. HoDoor Dataset

To the best of our knowledge, there are currently no available datasets that contain images of annotated doors of various opening types. To overcome this problem, we created a new dataset called the HoDoor (Hands on Door) dataset (The HoDoor dataset is available at https://puh.srce.hr/s/Y3BefR7N39r8Ccd, accessed on 24 January 2023).

The sensor used to capture the images is the Intel RealSense L515 LiDAR depth camera, which is capable of capturing 8-bit RGB and 16-bit depth images at a wide range (0.25–9 m) and high accuracy (on average < 14 mm @ 9 m). The maximum resolution of the RGB image is 1920 × 1080, while the depth images have a maximum resolution of 1024 × 768. The maximum frame rate of the camera is 60 FPS, enabling real-time acquisition of detailed RGB and depth data. The RGB camera has a field of view (FOV) of 69° × 42°, while the depth camera has an FOV of 70° × 55°. This allows the camera to capture both depth and RGB data over a wide area.

The RGB and the depth images were captured using the Robot Operating System (ROS) and saved as PNG files. The resolution of the images in the dataset is 640 × 480 pixels, and the RGB and depth images are aligned. A pair consisting of an RGB image with a corresponding depth image can be seen in Figure 2. Intrinsic camera parameters are included in the dataset. The dataset consists of three classes that differ in the way they are operated: push, pull, and handle. The push class describes doors without regular handles that need to be pressed on the front surface to open, while the pull class describes doors without regular handles that have some sort of indentation on one of the sides that provides space for fingers to grab and open the door. We also refer to the latter class as the hidden handle class. In contrast to the other two classes, the handle class represents room doors with a regular handle on the front. We divided the object instances into four categories of doors that can be opened: cabinet, closet, door, and drawer. Each object was captured from one or more viewpoints, but the number of viewpoints varies for each one. We also captured multiple images for each camera viewpoint. The first image represents the baseline image of the object without hands. The other images were taken from the same viewpoint but with hands grasping the interaction spots of the object. Interaction spots refer to spots on an object where the object can be opened. These can be a regular handle on the door, edges of the front surface of the door, or a part of the outer surface around which the push mechanism is located inside the door. The method of capturing the images with hands consisted of a person naturally and spontaneously touching or holding the interaction spots at various locations on the object, as if they were opening it on any given day. The information about the camera poses is not included in the dataset.

In addition, the class for each object was determined by opening the object. The Label Studio [63] data annotation platform was used for labeling. The images usually contain multiple doors, but only one was annotated, for which we use the term “relevant door”. The YOLO annotation format was used to export the bounding box annotations. For each image, a TXT file was created containing the object class, bounding box coordinates, and the height and width of the bounding box for the corresponding image. Figure 3 shows some examples of the annotations. The top row in the figure contains the baseline images with the corresponding bounding boxes, while the bottom row contains images taken from the same viewpoint but with a hand placed on interaction spots.

Figure 4 shows several examples of the images in the dataset described. The figure is divided into three parts, each containing an object from the three classes described above. For each class, two viewpoints of the object are shown in separate rows. The first image in the row is the baseline image, while the other images contain hands placed on the interaction spots. Since this is a real-world dataset, the images were taken at multiple locations, and their scenes vary in complexity. This scene complexity varies depending on the number of objects in the image, clutter, and occlusion. Details about the dataset are given in Appendix A.

#### Statistics

We captured a total of 2571 images representing 409 viewpoints with 224 different object instances. A comparison of the number of instances, viewpoints, and images per class is shown in Table 1. The push class had the lowest amount of data collected because it is less common in households compared to the other two classes. The largest amount of data was collected for the pull class, as handleless kitchen objects that can be opened by pulling on one of the sides are widely used today.

The distribution of object instances among categories can be seen in Figure 5. Of the total 224 object instances, 109 instances are in the cabinet category (1224 images), 101 in the drawer category (1208 images), and 13 in the closet category (124 images), while the door category, which includes room doors, has 1 instance (15 images). The closet and door categories have a small number of instances because there are not many instances that belong to the push or pull classes. However, cabinets and drawers are much easier to find in homes, and the various designs of these categories make it easier to capture a lot of diverse data. In addition, the object categories are not relevant to this research, as we only focus on the door opening types. Nevertheless, the information about the object categories is provided in the dataset for other potential users who might find it useful.

In Table 2, a distribution of the number of viewpoints across the three classes is shown. We mostly captured the objects from two viewpoints, as this covers most of the object’s environment, giving it enough context. If an object is placed in an area where it will be partially obscured or difficult to reach at most desired recording angles, that object will only be captured from one viewpoint angle. Objects were rarely captured from three viewpoint angles, mainly because the third angle does not provide much new information about the object and its environment. In these cases, the object’s surroundings do not change much compared to the first two viewpoints, especially when multiple objects of the same design are connected. In addition, sometimes the geometry of the room prevented us from capturing the third viewpoint of the object, whether it was a frontal viewpoint or a side viewpoint.

Figure 6 shows the distribution of the number of images in the viewpoints for each class. The images in the viewpoints include both baseline images and images with hands. While there is one viewpoint with 20 images in the pull class, most viewpoints in all classes have 5 to 8 images, which are sufficient to describe how to open the respective doors with the pose of the hand. The handle class has a lower median number of images in the viewpoints since the area of the handle is usually smaller than the area that can be used to open doors of other classes. This limits the number of possible hand poses on the handle.

### 3.2. Classification

In this paper, we present a method for classification of furniture door opening types, a diagram of which is shown in Figure 7. Even though our dataset consists of RGB and depth data, we utilize only the RGB data for this method. As previously mentioned, we consider three types of handling: regular handles (handle), hidden handles (pull), and push mechanisms (push). Our goal is to identify regions of RGB images representing doors or drawers of cabinets and classify these regions into one of these categories. We consider two types of images: those with a human hand demonstrating the opening type and those without. First, an object detection network is used to detect a region of interest (ROI) in a given image. This ROI is then *squarified* and, in the case of no human demonstration, fed to a classifier CNN to determine the class of the door or drawer. In the case with human demonstration, the squarified ROI is first fed to a hand pose detection network that outputs heatmaps—sometimes referred to as belief maps or confidence maps in the literature—for hand keypoints, and these heatmaps are fed to a classifier CNN to determine the class. While any object detection network can be used for our method, a hand pose detection network that outputs the aforementioned heatmaps is needed. For this purpose, we use OpenPose [1].

#### 3.2.1. *Squarification*

After an ROI has been selected, the ROI is squarified. This is done in order to decrease distortion of the input for the classification network, since the ROIs can have very different width-to-height ratios. Given a minimum size of an ROI, $dim_{min}$, and the input ROI’s width and height, a new dimension *w* is determined by selecting the largest among width, height, and $dim_{min}$ and increasing it by a given percentage, $dim_{increase}$. Given the center of the original ROI and *w*, a new ROI is determined using *w* as both its width and height. This ROI is then cropped so as to not fall outside of the original image. This may result in an ROI that is not an exact square but will still be close to a square, and the distortion as a result of resizing will decrease. A few examples of squarification are shown in Figure 8.

#### 3.2.2. ROI Classification

For classification without human demonstration, these extracted ROIs are then used as input for a CNN classifier. On the other hand, when classifying with human demonstration, we first feed these ROIs into OpenPose, which outputs heatmaps for human hand keypoints. The first 21 heatmaps are used for the 21 keypoints of a hand, where the peak of each heatmap predicts the position of the corresponding keypoint. The 22nd heatmap (#21) is a belief map for the background. An example of a well predicted and an example of a poorly predicted hand pose and a few of the corresponding heatmaps are shown in Figure 9. All of these heatmaps are then fed to a CNN classifier to obtain the handling type. We opt for using a hand pose detection network instead of using the RGB data as input for the classifier because our dataset is not very large and there are only a few different hands demonstrating the opening type. On the other hand, a network trained on a large dataset of hand poses should be more robust to images with different types of hands, environments, and conditions, thus making our method more robust as well. We believe the hand pose provides the necessary data for classification.

## 4. Results

We conducted two groups of experiments: one with images with human demonstration and the other with images without humans. For experiments, we use ground truth ROIs labeled by humans to better train the classifiers and to evaluate them separate from the detection.

### 4.1. Human Demonstration

First, the dataset is split into training (≈80%), validation (≈10%), and test (≈10%) sets. Since there are multiple images of each furniture from each viewpoint, images from the same viewpoint are not put in different partitions of the dataset. The dataset is split in such a way as to keep the distribution of images per class similar in each partition (Table 3). It is also partitioned in such a way as to keep the distribution of images per viewpoint angle (scene) similar. These distributions are shown in Figure 10.

As described in Section 3.2, the images are preprocessed by extracting a squarified ROI and then running it through OpenPose [1] (Instead of the code provided by the authors, we use the following pytorch implementation of the network: https://github.com/Hzzone/pytorch-openpose, accessed on 24 January 2023). The squarified ROI is fed to the network to obtain 22 heatmaps. These heatmaps are of size 92 ×N, where *N* depends on the original aspect ratio of the input ROI. The heatmaps are then resized to the size 92×92. Exactly because of this step the ROIs are squarified, so as to prevent high levels of distortion in the heatmaps. Resized heatmaps are then used for training ResNets [2].

For squarification of the input ROI, we set $dim_{min}$ to be 200, and we set $dim_{increase}$ to be from the set {0.0,0.2,0.5,0.8,1.0}. We choose this value for $dim_{min}$ to ensure that the hand is visible and can be detected in the picture. The $dim_{increase}$ is there to give more context for hand pose detection. We do not go over 100% increase since, even at this point, the OpenPose network starts detecting background as hand keypoints. We use 4 different ResNet structures: ResNet18, ResNet34, ResNet50, and ResNet152. We train each combination of network structure and $dim_{increase}$ value for 100 epochs with batch size of 32. The networks are trained on a single RTX 3080 GPU. We achieve the results shown in Table 4 for the validation set.

Based on the results from the validation set, the ResNet50 network was chosen with the $dim_{increase}$ parameter equal to 0.5 or 50%. Using this model, accuracy of the test set is 81.39%. The confusion matrix is shown in Figure 11.

### 4.2. Without Human Demonstration

Similarly, the dataset is split into training (≈80%), validation (≈10%), and test (≈10%) sets. There is only one image without human demonstration per viewpoint, but there are multiple viewpoints per furniture instance, so images of the same furniture instance are not put in different partitions. Similar to the partitions of the images with human demonstration, both the class distribution and the image per furniture instance distributions are similar. These are shown in Table 5.

The squarified ROIs are extracted and rescaled to size 128 × 128. Additionally, while training, we used the following random transformations on the input images:translation with maximum factor of 0.05,scaling with maximum factor of 0.05,rotation with maximum angle of 15°,color shift with maximum value change of 15 for each color,brightness change with maximum factor of 0.2, andcontrast change with maximum factor of 0.2.

The probability for each of these transformations is set to 0.5. Additionally, the input ROIs are normalized with mean and standard deviation of ImageNet [32] both for training and evaluation.

In this case, we do not train the networks from scratch. Instead, we use the networks pretrained on the ImageNet dataset [32]. We fine-tune ResNets on these ROIs using the same parameters as in Section 4.1. We achieve the results shown in Table 6 on the validation set.

Looking at these results, there are many models that achieve 100% accuracy on the validation set. Because of this, the ResNet50 network was chosen with the $dim_{increase}$ parameter equal to 0.5 or 50%, the same as in the experiment with human demonstration. Using this model, accuracy of the test set is 88.89%. The confusion matrix is shown in Figure 12.

#### Classification by Humans

To test whether the classes can be reasonably predicted using only RGB images, we created a survey that asked participants to classify 45 furniture instances with doors from the test set into one of the three classes. The test set is the same one used in the previous experiments. They were given the whole image without the ROI and the whole image with the ROI highlighted. A total of 69 people participated in the survey and achieved an accuracy of 73.66%. The confusion matrix for this experiment is shown in Figure 13. If, however, we consider the participants as voters and, for each image, we take their most voted answer as their prediction, they achieve an accuracy of 80%.

### 4.3. Detected ROIs

The aforementioned experiments are done on ground truth ROIs; however, we also perform experiments with ROI detection. For this purpose, we train YOLOv7 [12] on the DoorDetect dataset [34], since our dataset is not large enough to train a robust object detector. The network is trained with the hyperparameters, the same as in [12], except that we use batch size 8. The network is trained on a single RTX 3080 GPU. The results for the validation set are shown in Table 7. Ref. [34] achieved mAP of 0.45 on this dataset using YOLOv3, but it is unclear at what IoU this mAP is calculated.

Since we only label one furniture instance per image, we are only concerned with the recall on our dataset. We consider any bounding box classified as door, cabinet door, or fridge door with 0.5 IoU or higher, with our ground truth bounding box as a true positive. A few examples of ground truth ROIs and positive predicted ROIs are shown in Figure 14.

When evaluating the network on the whole dataset with human demonstration, we achieve recall of 88.58%. On the test set, we achieve recall of 85.71%. Furthermore, positive predictions are taken as the new ROIs and used as inputs for the classifier from Section 4.1. We achieve accuracy of 72.08%. Taking these two values, we can calculate the final accuracy on the test set to be 61.78%.

Similarly, we use the same pipeline for the dataset without human demonstration, where we achieve recall of 93.64%. On the test set, we achieve recall of 95.56%. The accuracy on the positive predicted ROIs is equal to 90.70%, which brings the final accuracy to 86.67%.

## 5. Discussion

Complex scenes can sometimes make some objects without regular handles difficult to classify, particularly when no hands are visible in the scene. The lighting in the scene can also affect the classification, as it may change the visibility of some visual features of the object. Figure 15 shows several images from the dataset that have proven difficult to classify for both humans and detectors. All of the objects in these images belong to the pull class but are often misclassified as push. The objects that are easily misclassified are usually captured from a perspective that does not show some important features of the object that would be critical for correct classification. In some cases, the objects do not even have visual features that distinguish them from another class.

Using our dataset with human demonstration, we show that a reliable classifier that relies exclusively on hand pose can be trained. The classifier’s performance depends on the underlying hand pose detection method [1], which brings additional uncertainty to the predictions. However, Figure 11 shows that most mistakes come from mistaking furniture doors with regular handles and furniture doors with hidden handles, which is to be expected since the hand gesture for opening those types of doors is somewhat similar. Furthermore, we show that the depth of the ResNet does not influence the results consistently, which is also to be expected, considering there are only three classes. What does consistently influence the results is the $dim_{increase}$ parameter, which controls how much context is included in the input for the hand pose detection. It can be seen that both too little context and too much context negatively influence the prediction.

It is hard to differentiate between doors with push mechanisms and those with a hidden handle because the main differences are a hidden mechanism or a differently designed edge. The former cannot be seen on a closed furniture door, and the latter is hard to notice on an RGB image. On the contrary, the hand poses used to open these types of doors are very distinguishable. Thus, it would be expected that a classifier would achieve better results on images with human demonstration than without.

However, the results from the dataset without human demonstration are different than what we expected. Using only the RGB information, we achieve perfect classification results on the validation set. On the test set, we achieve a very high accuracy of 88.89%. Compared to the results of the human survey (73.66%), this is suspiciously high. This is most likely due to the similarity between the dataset partitions. Since the dataset partitions contain images of similar furniture, the network overfits to those furniture instances, which brings the accuracy to such a high number. For future research, the dataset should be supplemented with additional images of furniture with doors that do not appear in the training set.

We also conducted experiments with detected ROIs. With these experiments, we show that a state-of-the-art network [12] trained on a different dataset [34], which contains images of furniture with doors, can reliably predict the ROIs. The results are better on the dataset without human demonstration, since there is no occlusion in form of human hands and arms. Furthermore, we show that the accuracy of our classification method on the images with human demonstration does slightly fall off. On the other hand, when considering images without human demonstration, the accuracy is similar for the ground truth ROIs and on the ROIs detected by the detector network. While our method can be used on scenes with multiple cabinets, drawers, etc., without human demonstration, our dataset is not labeled in such a way and thus we do not test it in this manner. When used on images with human demonstration and multiple cabinets, closets, or drawers, our method would need to be augmented with a discriminator that could differentiate between the ROIs with and without a human hand.

As mentioned earlier, the dataset should be supplemented with additional images of objects not appearing in the training set for the test set. With this new test set, more credible results could be obtained for classification without human demonstration. This classification could also be improved by using the depth information that is present in our dataset. Another future research option is robot manipulation of doors with push mechanisms and hidden handles, whereby our dataset could be used for teaching a robot how to handle these types of furniture doors based on human demonstration.

## Figures and Tables

**Figure 1 sensors-23-03093-f001:**
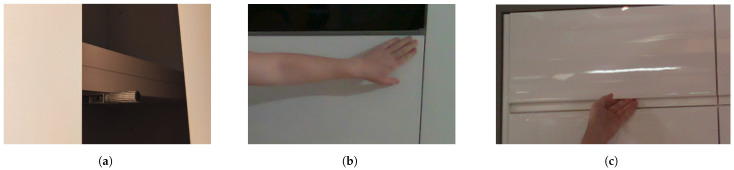
Opening doors without handles. (**a**) A push mechanism. (**b**) An example of opening of the push mechanism door. (**c**) An example of opening of the door with hidden handle.

**Figure 2 sensors-23-03093-f002:**
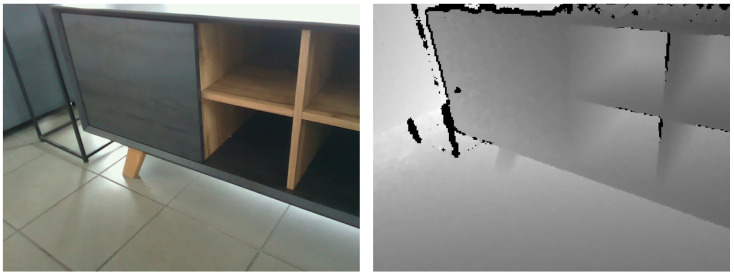
An example pair of an RGB image and a corresponding depth image contained in the dataset.

**Figure 3 sensors-23-03093-f003:**
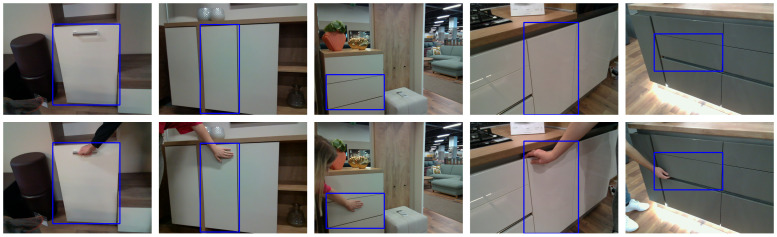
Sample images with annotations for the door of the object. **Top row**: Images without human demonstration. **Bottom row**: Corresponding images with human demonstration. The images in the bottom row were captured from the same viewpoint as the top row.

**Figure 4 sensors-23-03093-f004:**
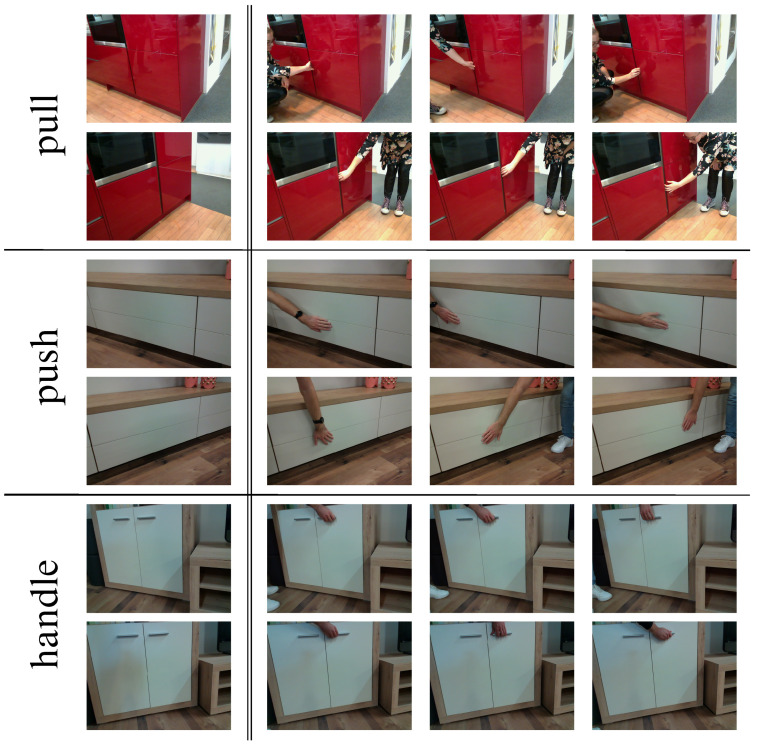
Sample images from our dataset representing the three types of handling considered. **Left**: Baseline images of the same object captured from different viewpoints. **Right**: Images with a human demonstrating the type of opening. The images in each row correspond with the baseline image on the left.

**Figure 5 sensors-23-03093-f005:**
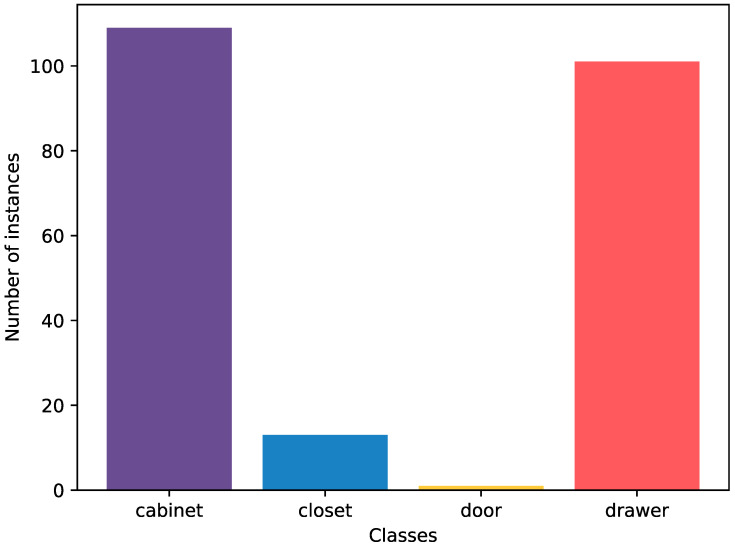
Distribution of object instances per category.

**Figure 6 sensors-23-03093-f006:**
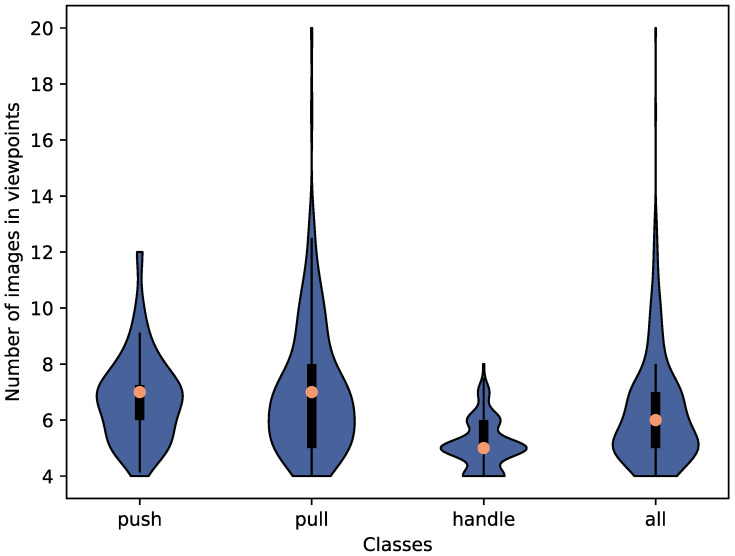
Distribution of images in the viewpoints per class and overall.

**Figure 7 sensors-23-03093-f007:**
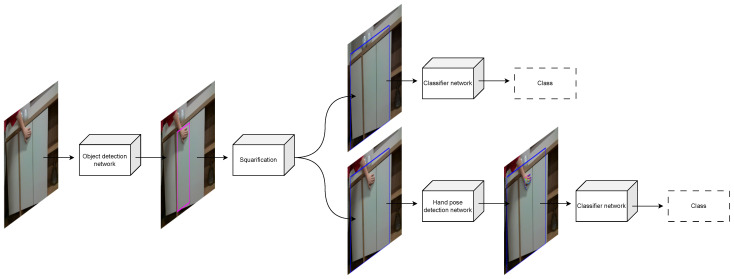
Diagram of the classification method. The top path shows the classification of images without a human hand demonstrating the opening type, while the bottom path shows classification of images with human demonstration.

**Figure 8 sensors-23-03093-f008:**
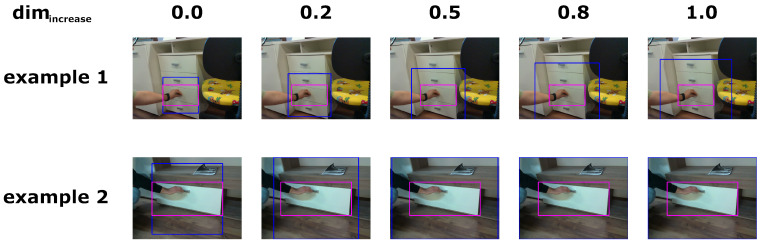
Examples of squarification with different values of $dim_{increase}$. The ground truth bounding boxes are shown in magenta, while the squarified bounding boxes are shown in blue.

**Figure 9 sensors-23-03093-f009:**
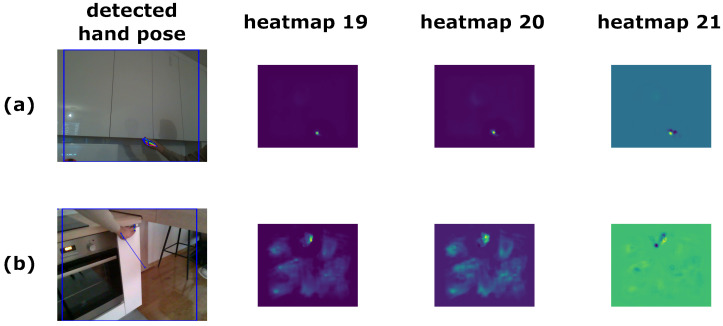
Two examples of results from hand pose detection using OpenPose. In (**a**), a well predicted hand pose is shown, while in (**b**), a poorly predicted hand pose is shown. When the predictions are good, heatmaps #0–20 have well defined keypoints.

**Figure 10 sensors-23-03093-f010:**
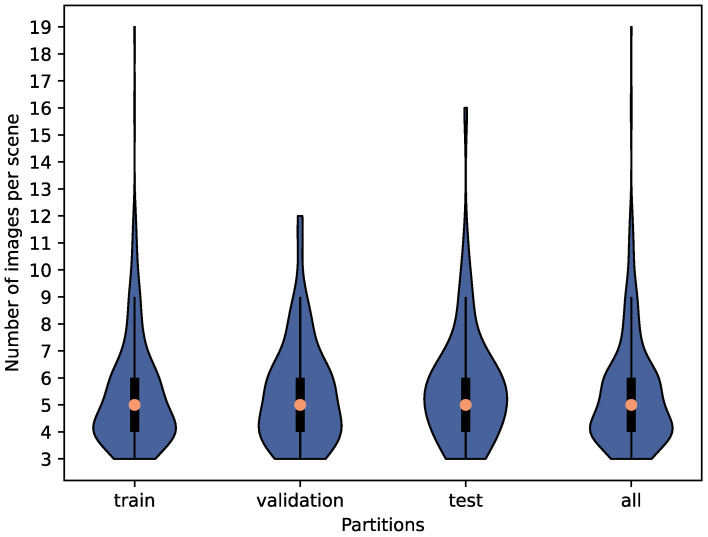
Distribution of images per viewpoint angle (scene) for each partition for images with human demonstration.

**Figure 11 sensors-23-03093-f011:**
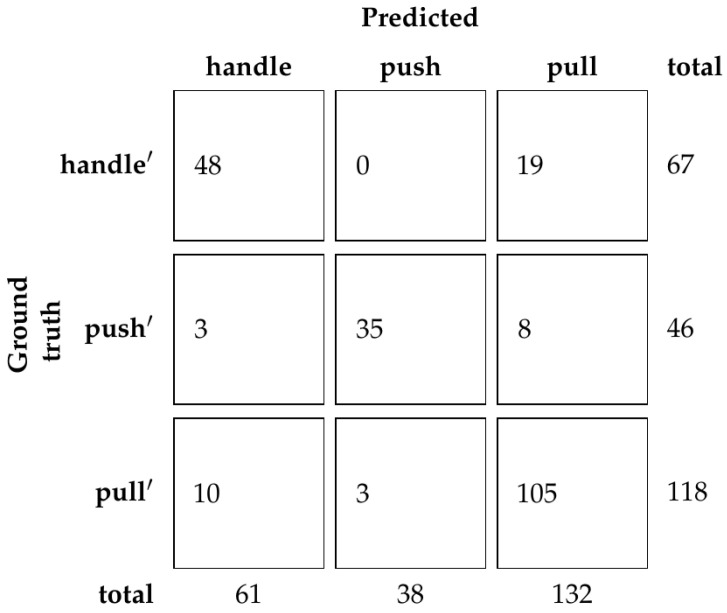
Confusion matrix of classification of the test set with human demonstration.

**Figure 12 sensors-23-03093-f012:**
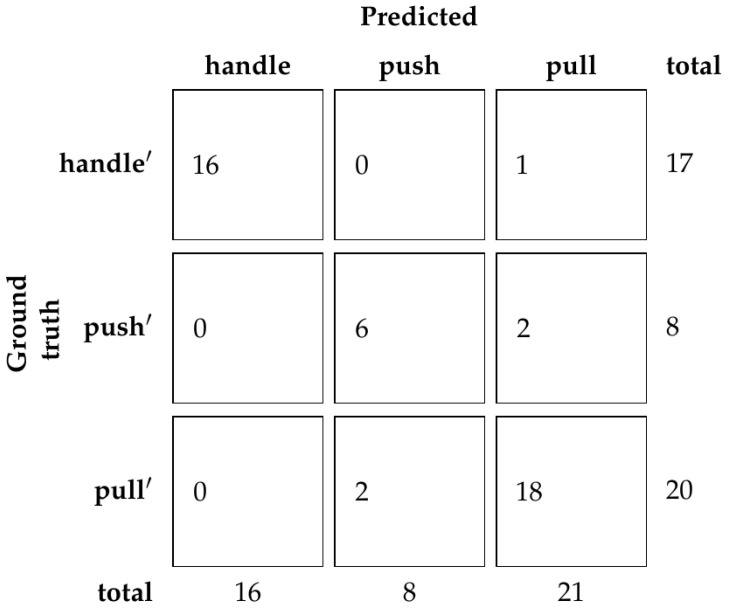
Confusion matrix of classification for the test set without human demonstration.

**Figure 13 sensors-23-03093-f013:**
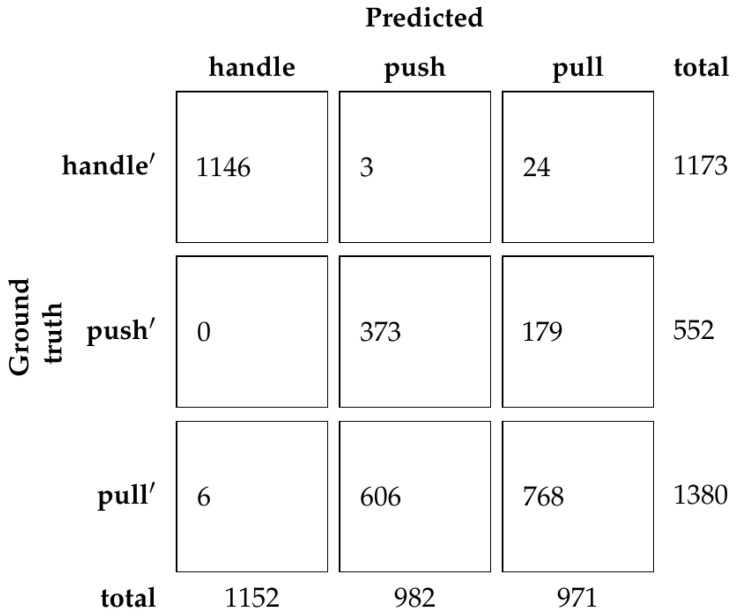
Confusion matrix for the human prediction survey on the dataset without human demonstration.

**Figure 14 sensors-23-03093-f014:**
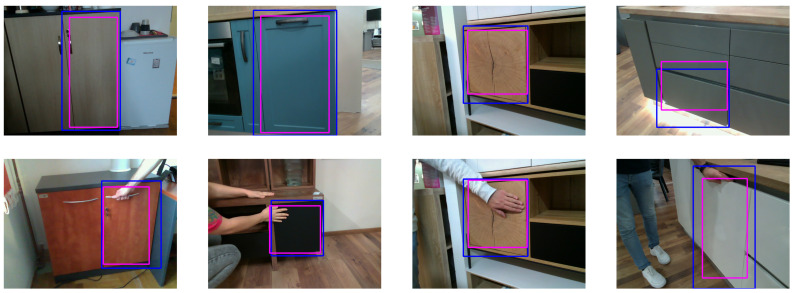
Examples of ROIs correctly predicted by YOLOv7. The ground truth bounding boxes are shown in blue, while the predicted bounding boxes are shown in magenta.

**Figure 15 sensors-23-03093-f015:**
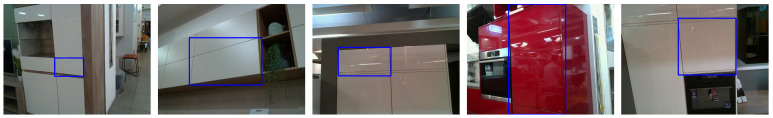
Several examples of objects that are difficult to classify due to their design and viewpoint angle of capture. All framed objects in the images belong to the pull class but were incorrectly classified as push.

**Table 1 sensors-23-03093-t001:** Distribution of object instances, viewpoints, and images per class.

	Push	Pull	Handle	All
Instances	43	104	77	224
Viewpoints	72	186	151	409
Images	489	1311	771	2571

**Table 2 sensors-23-03093-t002:** Distribution of viewpoints across classes.

	Class	Push	Pull	Handle
Viewpoints				
1		14	24	5
2		29	78	70
3		0	2	2

**Table 3 sensors-23-03093-t003:** Distribution of images per class for each partition for images with human demonstration.

	Push	Pull	Handle	All
Training	323 (18.92%)	893 (52.31%)	491 (28.76%)	1707 (100%)
Validation	48 (21.43%)	114 (50.89%)	62 (27.68%)	224 (100%)
Test	46 (19.91%)	118 (51.08%)	67 (29.00%)	231 (100%)

**Table 4 sensors-23-03093-t004:** Accuracy of the validation set with different combinations of network architecture and $dim_{increase}$ parameter for a dataset with human demonstration.

	*dim_increase_*	0.0	0.2	0.5	0.8	1.0
Network						
ResNet18		71.43%	69.20%	76.34%	79.91%	77.23%
ResNet34		69.64%	69.20%	78.12%	79.91%	72.77%
ResNet50		74.55%	78.12%	**81.25%**	79.02%	75.89%
ResNet152		75.89%	70.54%	77.23%	78.57%	76.34%

**Table 5 sensors-23-03093-t005:** Distribution of images per class for each partition for images without human demonstration.

	Push	Pull	Handle	All
Training	56 (17.50%)	147 (45.94%)	117 (36.56%)	320 (100%)
Validation	8 (18.18%)	19 (43.18%)	17 (38.64%)	44 (100%)
Test	8 (17.78%)	20 (44.44%)	17 (37.78%)	45 (100%)

**Table 6 sensors-23-03093-t006:** Accuracy of the validation set with different combinations of network architecture and $dim_{increase}$ parameters for the dataset without human demonstration.

	*dim_increase_*	0.0	0.2	0.5	0.8	1.0
Network						
ResNet18		100.0%	95.45%	100.0%	100.0%	100.0%
ResNet34		100.0%	100.0%	97.73%	100.0%	97.73%
ResNet50		90.91%	100.0%	100.0%	97.73%	100.0%
ResNet152		93.18%	95.45%	95.45%	97.73%	93.18%

**Table 7 sensors-23-03093-t007:** Results of object detection with YOLOv7 on the DoorDetect validation set.

Class	Labels	Precision	Recall	mAP@.5	mAP@.5:.95
door	73	0.755	0.548	0.624	0.379
handle	388	0.683	0.521	0.579	0.193
cabinet door	334	0.813	0.766	0.834	0.462
fridge door	106	0.84	0.604	0.683	0.472
all	901	0.772	0.609	0.68	0.376

## Data Availability

The HoDoor dataset can be found here: https://puh.srce.hr/s/Y3BefR7N39r8Ccd, accessed on 24 January 2023, and upon request to the authors by email.

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
