# Peer review of "Introduction to Door Opening Type Classification Based on Human Demonstration"

_sensors, 2023, doi:10.3390/s23063093_

Round 1

Reviewer 1 Report

This study describes a dataset of various types of door images and reports a preliminary study on classification using deep neural network models for these opening types. Door opening remains an important problem as a typical example of articulated objects manipulation by robots. In particular, how robots can know how to open a door without a handle is a difficult challenge. I think this paper can be useful as a clue to tackle this problem.

There are some comments on unclear points and suggestions for improvement:

1) Is the depth image not used in the experiments in this paper? How about introducing an example of a pair of an RGB image and a depth image, for example in 3.1?

2) p.11: Regarding the output heatmap of OpenPose, I understand each heatmap estimates the position of a specific part of the body, but what is the difference between heatmaps 19, 20, and 21 in Figure 8? Do all heatmaps represent hand positions? Please add some explanation.

Also, which heatmap did you use for the CNN model input?

3) I thought it was a good idea to prepare images with human demonstration to help to identify how to open the door, but contrary to my expectations, the case without human demonstration has a better discrimination rate for the opening type, as shown in 4.1 and 4.2.

This is mysterious and potentially misleading, so I would like you to elaborate on possible causes. Also, please explain the usefulness of images with hand demonstration. For example, for the “handle”, the case with human demonstration has a much lower success rate (Table 5). It may be that the door handle is not recognized well because the hand hides the handle. Also, as a future work, it would be interesting to compare the results with the same datasets used in the human prediction survey.

Reviewer 2 Report

The authors present an approach for identifying the different opening tecniques of doors, cabinets, and drawers through computer vision. In particular, they address the relevant issue of identifying what they call 'hidden handles' and push mechanisms that are very common in modern drawers and cabinets.

The reviewer appreciates the motivation and the state of the art reported by the authors, underlying the lack of relevant works to address the issue mentioned above. It is also apparent the effort of the authors in the dataset creation and labeling. In general, the paper is well written and clear, except some sentences.

However, in my opinion the main limit of the current version of the paper lies in the presented results.

Now I will go through the paper to provide my detailed comments:

- Although I appreciate the authors' idea of involving people for comparing the detectors results  with human-level identification capabilities, I believe that it cannot be stated among the contributions as it is too specific for the problem under investigation

- Lines 282-283 are not very clear. They seem to be too trivial and report a technical detail on the folders naming. Moreover it is unclear where the object directory is 'explained below'.

- In line 284 the authors refer to 'one relevant door per image'. What do they mean by relevant? Probably the authors should better clarify that their method is unable to perform multiple detection within a single frame and to state it as one of the limits of their work. Moreover, such an issue is rather relevant as, looking at Fig. 2, there may be several similar doors in a single frame. If the detector keeps jumping from one door to another, it may be extremely difficult to deploy it in a real world application including, e.g., a manipulator. A discussion by the authors on this point could be helpful.

- Lines 302-305: the unequal distribution of the instances among the classes is a bit controversial. Typically in datasets for classification through deep learning the main goal is to have all the classes equally represented, otherwise the deep neural network could tend to misclassify those classes or, even worse, to no classify them at all, as they would be just corner cases. The authors are encouraged to expand their dataset.

- Line 309: similarly to the comment above, the authors should definitely improve the distribution among categories represented in Fig. 4. Moreover, it is unclear to me the reason for the few examples for closets and doors provided in line 310, as the authors could have included more instances from other classes (e.g. handle).

- Among the Statistics reported in Sec. 3.1.1, if I am not wrong, it does not seem to be reported the number of images per category (only the number of instances is reported, that I assume is the number of objects, which are acquired from different viewpoints).

- A scheme showing where the 2 viewpoints are (as mentioned in line 314) could help the reader. Moreover it is unclear why a third viewpoint does not provide much new information as stated in line 319 as I would have expected to see at least a right, frontal, and left viewpoint.

- Sec. 3.2.: Why the authors classifed only cabine opening types? This seems to support that the closet and door categories were too under-represented in the dataset to be properly classified by the detector.

- Line 339: Why do the authors need heatmaps? It is clear only after reading the Results section.

- In Fig. 5 probably the authors forgot to include the unit for the number of viewpoints, otherwise the plot cannot be properly read.

- The authors should mention the rationale behind the choice of the dim_min and the dim_increase values.

- Lines 392-401: why do the authors included such a data augmentation for images without human demonstration only?

- As already mentioned above, I agree with the authors in lines 464-466 but they are encouraged to address it.

- I can't understand the sentence in lines 472-474.

- Finally it is unclear how the dataset with human demostration is supposed to provide better results in classification task. When the classifier is employed, obviously no human hand is visible in the scene. Could the authors discuss on this? Otherwise it seems unclear why they have introduced human demonstration as in the end they obtained worse results than without.

Minors:
- Probably 'pick' would be more appropriate than 'remove' in line 26
- To help the reader, I would suggest to change 'angle' with 'perspective, point of view, viewpoint' in line 73 and many other lines throughout the paper
- What do the authors mean by 'object' in line 105? Did they mean 'image' instead?
- In my opinion, the description of Yolo v7 (lines 105-120) should be shrinked as it is too detailed for being just a work in the Related Work
- Title of Sec. 3 should be 'Method' (line 247)
- In line 260, change 'Robotic Operating System' into 'Robot Operating System'
- In lines 271-273, it is unclear what the auhtors mean by poses
- In line 348: what the authors mean by 'to the sides of the image'? Moreover in line 349 the authors should better explain that the 'quasi-square' ROI occurs when the enlarged ROI falls outside of the original image
- In line 370 it is unclear why the authors specified "with human demonstration" as squarification of ROIs is performed in any case
- In my opinion Table 10 should be removed as neither Yolo v7 nor DoorDetect dataset have been developed by the authors.

Reviewer 3 Report

Authors are suggested to make some of the sections more concise and illustrate better their approach. How does the uneven data set affect quality? - needs to be elaborated better.

Good work with the Appendix information.

Round 2

Reviewer 2 Report

The authors addressed all the issues raised. Thank you.

Just a small remark on the following lines: (273-275)

"The number of viewpoints is not the same for each object, as is the pose with respect to the camera. Poses of the camera between different viewpoints are not included in the dataset. For each viewpoint of the object, we captured several images."

I do know what a pose is. What I meant is that the sentences above are a bit confusing. In fact, you first implicitly talk of the pose of the object "with respect to the camera", then you explicitly mention "poses of the camera", and finally you mention "viewpoint of the object". The concept of viewpoint is related to the camera rather than the object. So at least the last sentence should be rephrased accordingly. Finally I would suggest to remove  "with respect to" and rephrase as "as is the pose of the camera".

Author Response

We wish to thank the reviewer for their constructive suggestions. 

We changed the pointed out sentence, and we believe it is now more clear:

" Each object was captured from one or more viewpoints, but the number of viewpoints varies for each one. We also captured multiple images for each camera viewpoint.....

....

The information about the camera poses is not included in the dataset."